# Control of Postharvest Gray Mold at Strawberry Fruits Caused by *Botrytis cinerea* and Improving Fruit Storability through *Origanum onites* L. and *Ziziphora clinopodioides* L. Volatile Essential Oils

İbrahim Kahramanoğlu [1], Olga Panfilova [2,*], Tuba Genç Kesimci [3], Ayse Usanmaz Bozhüyük [3], Ramazan Gürbüz [3] and Harun Alptekin [3]

[1] Department of Horticulture, Faculty of Agricultural Sciences and Technologies, European University of Lefke, Gemikonagi, Northern Cyprus, Mersin 99780, Turkey; ibrahimcy84@yahoo.com
[2] Russian Research Institute of Fruit Crop Breeding (VNIISPK), Zhilina 302530, Orel District, Orel Region, Russia
[3] Department of Plant Protection, Faculty of Agriculture, Iğdır University, Iğdır 76000, Turkey; tuba.genc@igdir.edu.tr (T.G.K.); ayseusanmaz@hotmail.com (A.U.B.); r_grbz@yahoo.com (R.G.); harunalptekinn04@gmail.com (H.A.)
\* Correspondence: us@vniispk.ru

**Abstract:** The present research was undertaken to study the antifungal activities of *Origanum onites* L. and *Ziziphora clinopodioides* L. essential oils against three different isolates (M1-5, M2-1 and M3-5) of *Botrytis cinerea* (in vitro tests) and to investigate the vapor contact impacts on fungus and strawberry fruit quality (in vivo tests). Antifungal activities of these oils were tested by following the poisoned food technique at four different concentrations (0.25, 0.50, 1.00 and 2.00 mL/L) against *B. cinerea*. In vitro studies suggested that the 0.50 mL/L and 1.00 mL/L doses of *O. onites* and 1.00 mL/L and 2.00 mL/L doses of *Z. clinopodioides* provide high mycelial growth inhibition, 85.29–94.12% and 39.12–94.12%, respectively, by direct addition to food. Thus, these doses were tested in in vivo conditions, as a vapor contact treatment against two isolates (M1-5 and M3-5) of *B. cinerea* inoculated on strawberry cv. Camarosa fruits. Results showed that both *O. onites* and *Z. clinopodioides* essential oils have a moderate to high impact on the prevention of gray mold. The oils were also found to have a slight to moderate impact on weight loss and the loss of soluble solids concentration. Overall, the results demonstrated that the tested oils are a potential biodegradable alternative to fungicides.

**Keywords:** *Botrytis* isolates; inhibition of mycelial growth; in vitro antifungal assay; in vivo postharvest studies; physical quality

## 1. Introduction

Postharvest quality of fresh horticultural crops is significantly affected by several biological factors, including respiration, transpiration, ethylene and phytopathogens and these factors can be managed by several environmental factors, such as temperature, relative humidity, air composition, light, heat, ethylene and biomaterials. Postharvest deterioration includes physiological changes, physical losses, bio-chemical changes and pathological deterioration [1,2]. Postharvest decay caused by *Botrytis cinerea* (gray mold) is among the main causes of the postharvest losses. This necrotrophic and airborne fungus attacks more than 200 crops, mostly including pome fruits, stone fruits, grapes and berries. The disease symptoms vary widely, but most of them include soft rots accompanied with water-soaked parenchyma tissues and followed by gray masses of conidia [3]. These symptoms decay the fruits and render their marketability. These infections damage the storability of the fruits and adversely affect the marketability of the fruits [4]. In addition to this, the secondary metabolites of the mold (mycotoxins) have direct adverse impacts on human health [5].

One of the most widely used techniques against the fungus is fungicides [6] which include the active ingredients of thiram, mancozeb and captan. However, excessive- and mis-use of fungicides have several disadvantages, including hazards on human health due to residues, negative impacts on non-target organisms, environment and development of resistant genotypes [7,8]. These impacts caused a decrease in the acceptability of fungicides with increased consumer awareness about agro-chemicals, and numerous researches have been conducted in order to identify ecological alternatives [9].

Heat (water or air) treatments [10], light irradiation [11], modified atmosphere packaging [12] and coatings with plant natural products [13] are among the most widely tested and used alternatives to fungicides. All of these methods have also significant advantages in terms of fruit preservation, including the prevention of weight loss, physiological and bio-chemical changes. Bio-based, plant natural products, including essential oils, have been widely tested in different fruits and are considered to be highly acceptable due to their highly biodegradable nature [14]. Essential oils may have the ability to induce some biochemical reactions in pathogens, control them, improve fruits' tolerance to pathogens by inducing biosynthesis of some protective phytochemicals, retard ethylene production in fruits, reduce the activity of free radicals, oxidation and prevent enzymatic degradation [15–17].

Strawberry (*Fragaria x ananassa* Duch.) fruits are the most popular berries because of their unique flavor and diverse and abundant phytochemical contents [18]. Strawberry fruits are very sensitive to postharvest storage because of their high respiration rate and physical characteristics. They are also very susceptible to phytopathogens where the infections are mainly caused by *B. cinerea* [19]. Plant essential oils, such as lemongrass oil, orange oil and mandarin oil, were reported as successful in controlling *B. cinerea* in strawberry fruits [20]. Besides this, yeasts also have a high potential to be used against phytopathogenic fungi, including *Botrytis cinerea* [21]. Antifungal activities of essential oils have been associated with several biochemical compounds, mostly including limonene, γ-terpinene, eucalyptol, pulegone and thymol [22–26]. *Origanum onites* L. and *Ziziphora clinopodioides* L. are two important species of the Mediterranean climate and their oils have been reported to have abundant contents of antifungal compounds. According to Kotan et al. [27], the major constituents of the *O. onites* essential oil are carvacrol (70.50%), *p*-cymene (13.97%) and thymol (2.19%). Moreover, *Z. clinopodioides* is rich in pulegone (29.31%), menthone (21.79%) and 1,8-cineole (eucalyptol 15.31%) [28]. Although there are several published studies about the antifungal and antibacterial activities of these two essential oils [26,29–31], the impacts of these oils on the *B. cinerea* have not been studied well and no information exists about their efficacy in relation to the storability of fruits. Therefore, the present study was conducted to determine the efficacy of *O. onites* and *Z. clinopodioides* on the *B. cinerea* in in vitro studies and to test their effects on the storability of strawberry fruits cv. Camarosa under in vivo conditions.

## 2. Materials and Methods

### 2.1. Plant Material and Isolation of Essential Oils

The aerial parts of the *Origanum onites* L. (Lamiaceae) and *Ziziphora clinopodioides* L. (Lamiaceae) plants were collected at the flowering stages (BBCH-scale: 65) from the Babadağ area of Fethiye/Muğla and Çiçekli village of Tuzluca/Iğdır, respectively, in Turkey between June and July 2020. Plant materials were brought to the Plant Protection Laboratory of the Iğdır University, Turkey. Aerial parts of the plants were dried in shade and were ground in a grinder. The dried plant samples (500 g) were subjected to hydro distillation for 3–4 h using a Clevenger-type apparatus. The oil yields of *O. onites* and *Z. clinopodioides* were found as 1.50% and 1.56% (*w/w*, dry weight basis), respectively. The essential oils (EOs) were stored in a refrigerator at 4 °C until they were used in the below described studies. The chemical compositions of the EOs were not tested in the present study.

### 2.2. In Vitro Antifungal Assays

#### 2.2.1. Fungi (*Botrytis cinerea*)

The three isolates of *B. cinerea*, M1-5, M2-1 and M3-5, were supplied from the culture collection of Iğdır University, Faculty of Agriculture, Department of Plant Protection and Phytopathology Laboratory. The isolates were identified microscopically according to the features of mycelium, such as appearance and color, as well as conidia, conidiophore and sclerotia. M2-1 coded isolate was selected from the three isolates, and DNA sequencing was performed by sequencing. The fungi cultures were kept and grown on Potato Dextrose Agar (PDA) slants at 25 °C for seven days.

#### 2.2.2. Antifungal Studies

Antifungal activities of the essential oils of *O. onites* (OO) and *Z. clinopodioides* (ZC) were tested with the poisoned food technique, as suggested by several previous studies [32,33], with slight modifications. Four different concentrations of both oils (0.25, 0.50, 1.00 and 2.00 mL/L) were prepared by adding the required amount of essential oils containing 1:2 (*v*/*v*) of Ethanol (70%) [20] to cooled (45 °C) molten PDA and rotating in a sterile Erlenmeyer flask. A total of 20 mL solution was added into each sterile petri dishes (9 cm in diameter). The medium was solidified at room temperature (23 ± 2 °C) for a duration of nearly 1 h. Then, agar discs with mycelia (5 mm diameter) of *B. cinerea* isolates (M1-5, M2-1 and M3-5) were cut from the periphery of actively growing regions of the seven day old isolate cultures and placed in the center of each plate. Moreover, three controls (1: without oil, 2: with 70% ethanol and 3: fungicide [500 g/L Fenhexamid (Teldor® SC 500, Bayer AG, Leverkusen, Germany) with an application dose of 100 mL/100 L]) were included in the assays. Control plates were also inoculated by following the same procedure. The plates were incubated at 25 °C [34] for seven days. Four plates were used per treatment. The colony diameters were read after three, five and seven days of incubation. The percentage of mycelial growth inhibition (colony diameter, measured in cm) was calculated according to the formula suggested by Thomidis and Filotheou [34]. Mycelial growth inhibition (%) = $\{[(dc - dt)/dc] \times 100\}$, where dc is the mean radial diameter of the pathogen in the control sample, and dt is the mean radial diameter of the pathogen in the treated sample.

### 2.3. In Vivo Postharvest Studies

#### 2.3.1. Fruit Materials

Strawberry cv. Camarosa fruits, at eating maturity (SSC: 7.0; pH: 3.1, BBCH-scale: 87), were taken from a commercial strawberry business. During budding and flowering, nitrogen, potash and phosphorus fertilizers were used. During the fruiting period, mineral fertilizers were not used. Pesticides have not been used. Fruits were harvested during the morning hours and quickly transferred to the Plant Protection Laboratory of Iğdır University, within 2 h under refrigerated (3.5 °C ± 0.5 °C) conditions.

#### 2.3.2. In Vivo Experiments and Antifungal Activity Assay

Vapor assay was used to determine the antifungal activity of the essential oils on the strawberry fruits. The doses of the oils and *B. cinerea* isolates were determined according to the results of the in vitro assay. In these in vivo studies, the 0.50 mL/L and 1.00 mL/L doses of *O. onites* and 1.00 mL/L and 2.00 mL/L doses of *Z. clinopodioides*, which were found to be more effective were selected and tested. Moreover, the studies were continued with two isolates of *B. cinerea*, the M1-5 and M3-5. First of all, healthy strawberry fruits visually selected by appearance and were disinfected in sodium hypochlorite at 2.0% for 5 min. Then, the fruits were cleaned under pure water three times and dried on sterile drying papers for 20–25 min. Hereafter, each fruit was wounded (1 mm wide and deep) using a sterile pipet tip (10 μL). A conidial suspension of *B. cinerea* isolates (10 μL) at concentration of $1.0 \times 10^6$ conidia/mL was then injected into each wound. After inoculation, four fruits for each replication were placed in PVC egg boxes with lid (dimensions of $7 \times 10 \times 15$ cm).

The selected doses from the in vitro studies (0.50 mL/L and 1.00 mL/L of air for *O. onites* and 1.00 mL/L and 2.00 mL/L of air for *Z. clinopodioides*) were used in current study. The boxes had a total volume of 600 mL which provides ~500 mL air space after putting fruits in. The application doses were calculated on the base of air space and these doses of essential oils were then soaked to a filter paper (20 mm$^2$) and placed onto the inner part of the box cover. To prevent the loss of volatile compounds, the plastic boxes were quickly sealed with parafilm (9 mic). Besides the essential oil treatments, three separate controls were included in the studies: (i) a control with sterile water soaking after artificial infection, (ii) a control with natural infection without any treatment, and (iii) a third control with fungicide (500 g/L Fenhexamid (Teldor® SC 500) with an application dose of 100 mL/100 L) application. The fungicide application was performed by direct spraying. Each treatment included three replications, each containing four fruits. Contrary to the regular procedures for storing fruits at 25 °C for seven days [35–37], to determine the effects under real storage conditions for strawberries, the fruit boxes were stored at 3.5 °C $\pm$ 0.5 °C [38] and 90–95% relative humidity for 21 days. Current research has focused on testing the effects of *B. cinerea* contact steam essential oil on strawberries, not just *B. cinerea*. This is why the fruits were stored at the regular/recommended storage conditions. The impacts of EOs on the *B. cinerea* (under favorable conditions of the fungi) was tested in the first part of the research in in vitro studies. Before storage, strawberry fruit were kept at 25 °C for 2 h to initiate infection [39]. The fungal infections and other quality parameters were measured at an interval of three days. At each measurement point, the disease severity (DS) of each fruit was observed according to the 0–5 scale of Huang et al. [40] where 0 represents no infection, 1 represents a less than 20% rotted area, 2 represents a 20.1% to 40% rotted area, 3 represents a 40.1% to 60% rotted area, 4 represents a 60.1% to 80% rotted area and 5 represents a more than 80.1% rotted area.

### 2.3.3. Fruit Quality Analysis

Besides the disease severity, weight loss, soluble solids concentration, pH and ascorbic acid content of the fruits were regularly determined for 21 days with three days' interval. The initial and final weight of the fruits were measured with a digital scale (sensitive to 0.01 g) and the percentage of weight loss was determined by following the standard ratio method. Digital table refractometer (WAY-2S, Seoul, South Korea) was then used to measure the soluble solids concentration (%) of the fruits. The pH of each fruit juice was then determined with a pH meter (Jenco Instruments Inc., San Diego, CA, USA). Finally, the amount of ascorbic acid (AA: mg 100 g$^{-1}$) content of the fruits was then assessed by titrating the fruit juice with iodine solution [41]).

### 2.4. Data Analysis

Microsoft Excel was used to calculate the mean and standard deviation of each treatment for each quality parameter and to prepare tables and figures from the raw data. The raw data of each parameter was also subjected to Analysis of Variance (ANOVA) separately for each measurement point by SPSS 22.0 and the statistical separation of the means were carried according to the Tukey's HSD test at a 5% significance level.

## 3. Results

### 3.1. Antifungal Activity of Essential Oils

The effects of different application doses of *O. onites* (OO) and *Z. clinopodioides* (ZC) essential oils by the direct contact method at PDA on the mycelial growth (colony diameter—cm) of three different *B. cinerea* isolates (M1-5, M2-1 and M3-5) during seven days of incubation are shown in Table 1. The appearance of colonies at day seven were also shown in Figure 1A,B. The highest mycelial growth of all isolates were obtained from the two controls (sterile water and 70% ethanol) after seven days of incubation. This made it possible to conclude that 70% ethanol had no significant influence on the mycelial growth. Different doses of the EOs were then noted to have variable impacts on mycelial growth.

The two highest doses of OO (1.00 and 2.00 mL/L) were observed to prevent the mycelial growth of all isolates of *B. cinerea*. Moreover 0.50 mL/L dose of OO also provided a significantly similar effect on the mycelial growth of M2-1 and M3-5 isolates. The lowest dose of OO provided less effect than the highest doses, but was higher than the two controls. The ZC was then noted to have less effect than the OO essential oil. Only the highest dose of ZC (2.00 mL/L) provided high performance in the prevention of mycelial growth. The second highest dose of ZC (1.00 mL/L) was noted to provide significantly less effect than the highest dose, but was higher compared with other doses and control treatments. The M2-1 isolate was found to be slightly more sensitive than the other two isolates.

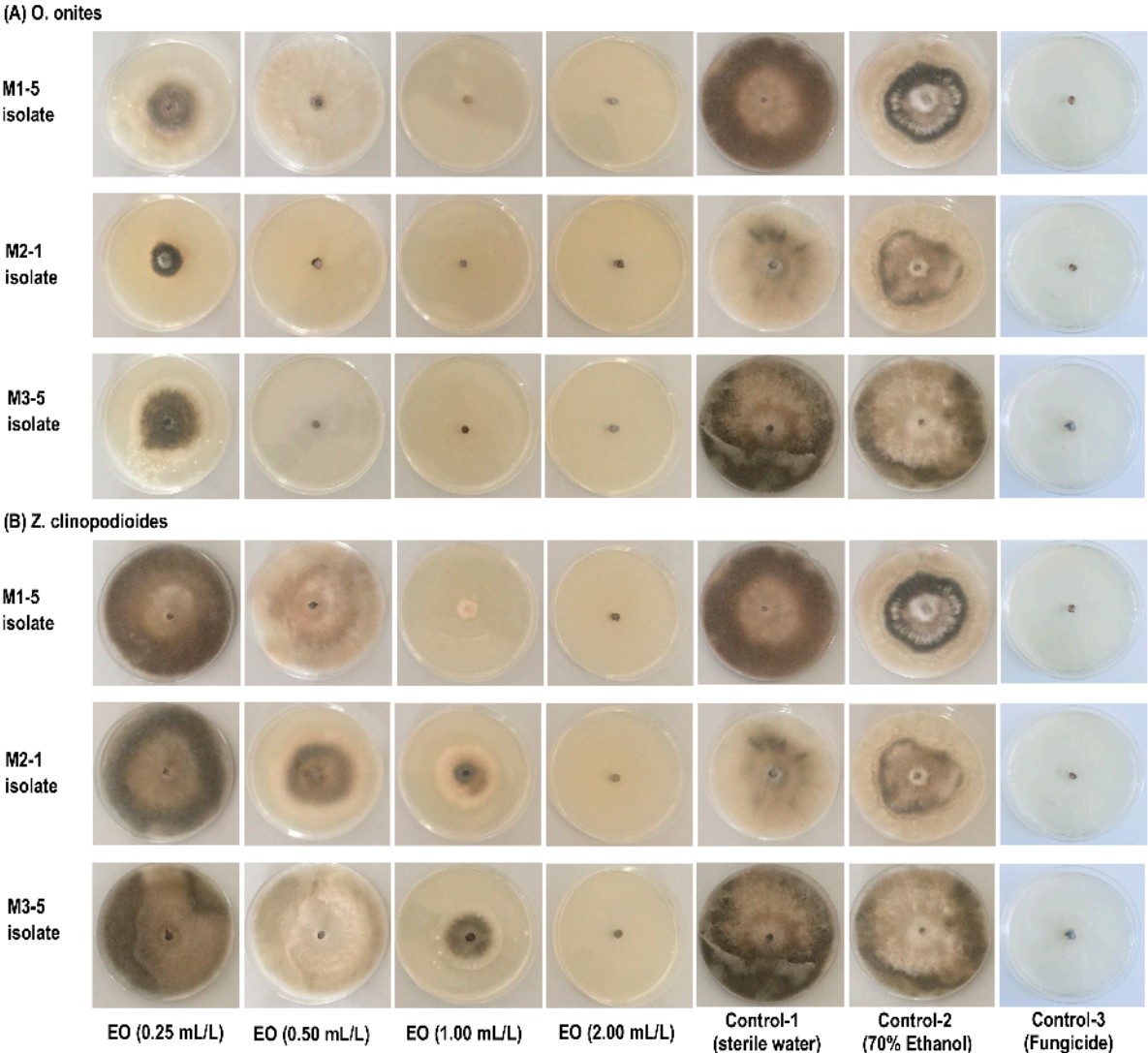

**Figure 1.** Effect of different application doses of (**A**) *O. onites* and (**B**) *Z. clinopodioides* essential oils, added to the PDA media, on the mycelial growth of *B. cinerea* isolates (M1-5, M2-1 and M3-5) after 7 days of incubation.

The mycelial growth inhibition (%) results are shown in Figure 2. According to the results obtained, the greatest antifungal activity (94.12%) against three isolates of *B. cinerea* was observed from the OO essential oil when applied in 1.00 and 2.00 mL/L doses. Moreover, the 0.50 mL/L dose of OO provided 94.12%, 91.18% and 85.29% mycelial growth inhibition on the M3-5, M2-1 and M1-5 isolates, respectively. The essential oil of ZC showed less potential for the control of three isolates of *B. cinerea* (mainly at lower doses), since the highest inhibition of mycelial growth was occurred at the highest dose of application.

**Table 1.** Effect of different application doses of *O. onites* and *Z. clinopodioides* essential oils, added to the PDA media, on the mycelial growth (colony diameter—cm) of *B. cinerea* isolates (M1-5, M2-1 and M3-5) during seven days of incubation.

| Treatments | 3 Days | 5 Days | 7 Days | 3 Days | 5 Days | 7 Days |
|---|---|---|---|---|---|---|
| | *O. onites* on M1-5 Isolate | | | *Z. clinopodioides* on M1-5 Isolate | | |
| EO (0.25 mL/L) | 1.53 d | 2.86 b | 4.09 b | 2.90 c | 7.75 ab | 8.50 a |
| EO (0.50 mL/L) | 1.70 c | 1.25 c | 1.25 c | 2.24 c | 7.00 b | 8.50 a |
| EO (1.00 mL/L) | 0.50 d | 0.50 d | 0.50 d | 0.60 d | 1.28 c | 2.56 b |
| EO (2.00 mL/L) | 0.50 d | 0.50 d | 0.50 d | 0.50 d | 0.50 c | 0.50 c |
| Control-1 (sterile water) | 5.41 a | 8.50 a | 8.50 a | 5.41 a | 8.50 a | 8.50 a |
| Control-2 (70% ethanol) | 3.90 b | 7.58 a | 8.50 a | 3.90 b | 7.60 ab | 8.50 a |
| Control-3 (fungicide) | 0.50 d | 0.50 d | 0.50 d | 0.50 d | 0.50 c | 0.50 c |
| | *O. onites* on M2-1 isolate | | | *Z. clinopodioides* on M2-1 isolate | | |
| EO (0.25 mL/L) | 0.79 b | 2.34 b | 3.19 b | 3.29 b | 7.95 a | 8.33 a |
| EO (0.50 mL/L) | 0.50 b | 0.50 c | 0.75 c | 1.76 c | 5.23 b | 6.86 b |
| EO (1.00 mL/L) | 0.50 b | 0.50 c | 0.50 c | 0.55 cd | 2.10 c | 5.18 c |
| EO (2.00 mL/L) | 0.50 b | 0.50 c | 0.50 c | 0.50 d | 0.50 c | 0.50 d |
| Control-1 (sterile water) | 4.49 a | 7.33 a | 8.50 a | 4.49 ab | 7.33 a | 8.50 a |
| Control-2 (70% ethanol) | 4.56 a | 7.99 a | 8.50 a | 4.56 a | 7.99 a | 8.50 a |
| Control-3 (fungicide) | 0.50 b | 0.50 c | 0.50 c | 0.50 d | 0.50 c | 0.50 d |
| | *O. onites* on M3-5 isolate | | | *Z. clinopodioides* on M3-5 isolate | | |
| EO (0.25 mL/L) | 0.99 b | 2.83 b | 4.50 b | 4.65 b | 8.23 a | 8.50 a |
| EO (0.50 mL/L) | 0.50 b | 0.50 c | 0.50 c | 3.14 c | 7.48 a | 8.50 a |
| EO (1.00 mL/L) | 0.50 b | 0.50 c | 0.50 c | 0.66 d | 2.16 b | 4.19 b |
| EO (2.00 mL/L) | 0.50 b | 0.50 c | 0.50 c | 0.50 d | 0.50 c | 0.50 c |
| Control-1 (sterile water) | 5.68 a | 8.50 a | 8.50 a | 5.68 ab | 8.50 a | 8.50 a |
| Control-2 (70% ethanol) | 6.10 a | 8.50 a | 8.50 a | 6.10 a | 8.50 a | 8.50 a |
| Control-3 (fungicide) | 0.50 b | 0.50 c | 0.50 c | 0.50 d | 0.50 c | 0.50 c |

Different letters next to the values at each incubation day for different oils and isolates represents significant difference according to Tukey's HSD test ($p \leq 0.05$).

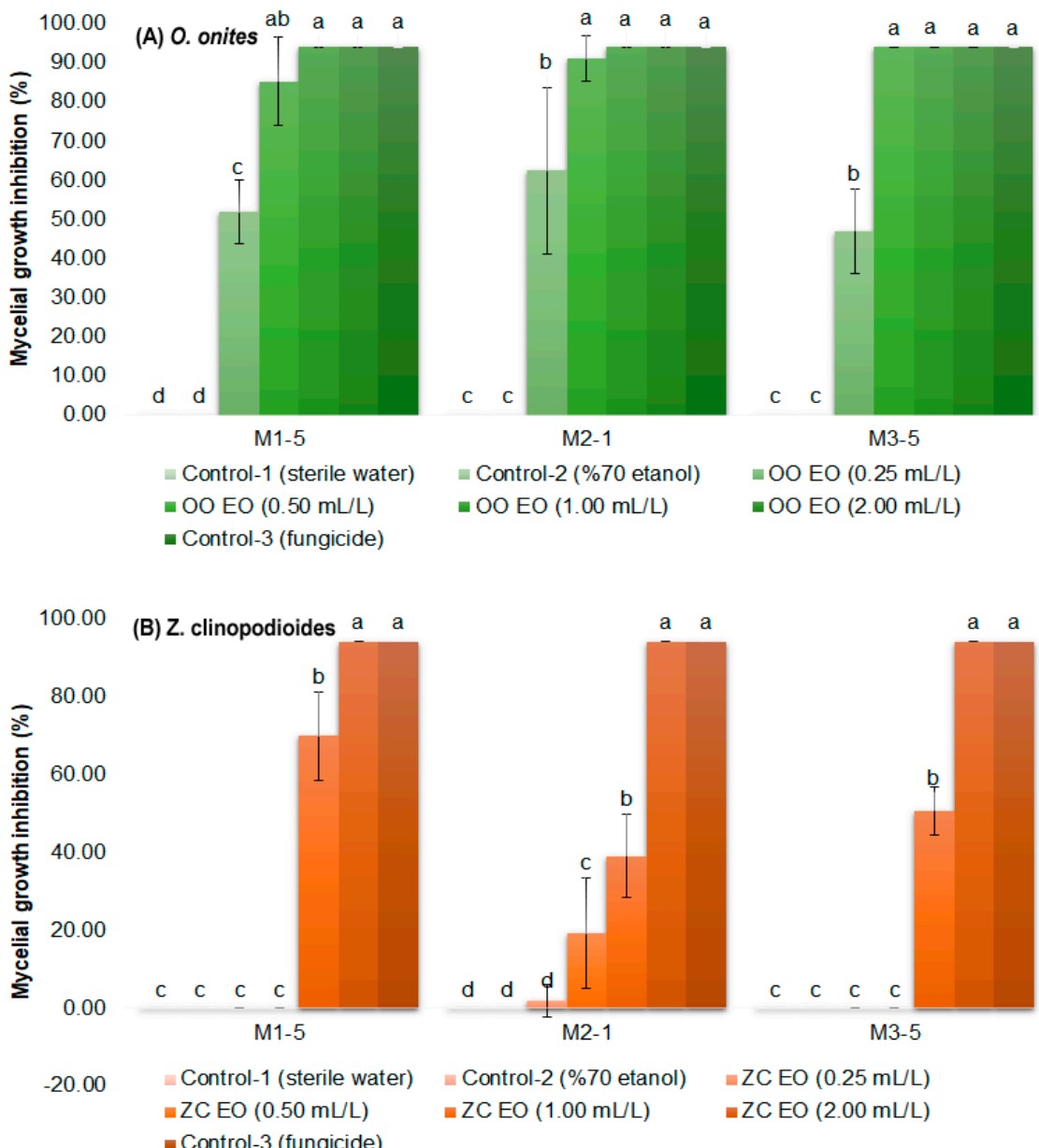

**Figure 2.** Inhibitory effect of different application doses of (**A**) *O. onites* and (**B**) *Z. clinopodioides* essential oils on the mycelial growth of *B. cinerea* isolates (M1-5, M2-1 and M3-5) on PDA based on contact assays after 7 days of incubation. Different letters next to the values of different isolates on the same rows represents significant difference according to Tukey's HSD test ($p \leq 0.05$).

*3.2. Impacts of Volatile Essential Oils on Disease Severity at Strawberry Fruits*

The impacts of vapor contact of *O. onites* and *Z. clinopodioides* essential oils on the disease severity of strawberry fruits infected by two isolates *B. cinerea* are presented in Figure 3. The statistical analysis revealed that the both of the EOs have a significantly higher impact than the artificially infected and sterile water soaked treatment, but a lower impact than the fungicide application. At the same time, the fruits with natural infection were also observed to have very high disease severity, which was significantly similar to the artificially infected fruits. Thus, it is clear that the EOs have a high performance in the prevention of disease severity. The higher doses of the EOs provided higher efficacy in reducing the disease severity of the fruits. Similar results were obtained for both of the fungus isolates. After the fungicide application, 2.00 mL/L of air ZC application was found

to be the most effective treatment in reducing disease severity. This treatment reduced the disease severity caused by the M1-5 isolate from 5.00 to 3.00 and by M3-5 isolate from 5.00 to 2.30 according to the 0–5 scale. As compared with the disease severity of fungicide treated fruits, 1.80 for M1-5 isolate and 1.50 for M3-5 isolate, both of the EOs can be accepted as highly effective in reducing the disease severity. Both of the EOs, but mostly ZC, were not only beneficial in reducing the disease severity at the strawberry fruits, but also in delaying the onset of disease during storage.

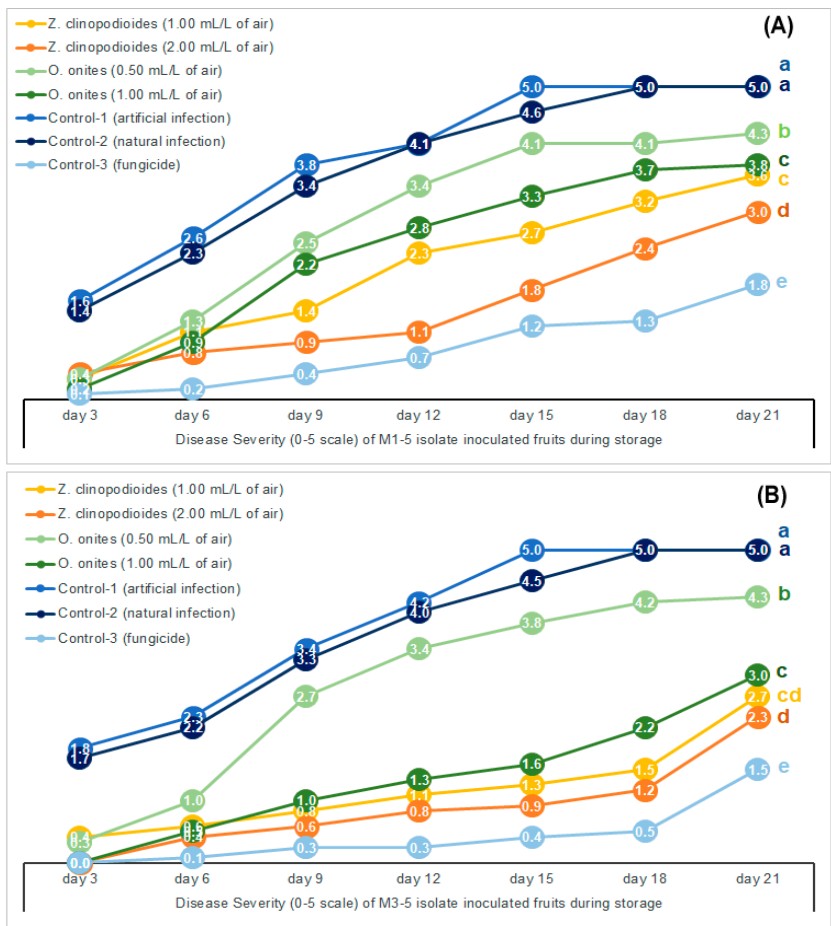

**Figure 3.** Vapor contact impacts of *O. onites* and *Z. clinopodioides* essential oils on the disease severity (0–5 scale) of (**A**) M1-5 and (**B**) M3-5 isolates of *B. cinerea* inoculated on strawberry fruits during 21 days of storage. Different letters next to the lines of each treatment represents significant difference at the 21st day of storage, according to Tukey's HSD test ($p \leq 0.05$).

### 3.3. Impacts of Volatile Essential Oils on Strawberry Fruit Quality

In addition to the antifungal activity, the impacts of the OO and ZC EOs on weight loss, ascorbic acid (AA) content, soluble solids content (SSC) and the pH of strawberry fruits were also examined and presented in Figure 4. Weight loss results were found to have a parallel trend with the antifungal activity of the treatments (Figure 4A). All EOs and fungicide treatment was found to prevent weight loss of the fruits. The higher application doses of the EOs were found to a have higher influence on the prevention of weight loss. Among the tested treatments, the highest efficacy was noted for the 1.00 mL/L of air application dose of OO. The results were similar for both isolates of *B. cinerea*. At the end of the storage period, no significant difference was observed among the different treatments for both fungi isolates. Although there is not any significant difference, the AA contents of the EO treated fruits were slightly less than the control fruits (Figure 4B). The SSC contents of the fruits had a declining trend during storage. This decline was slightly prevented at the EO treated fruits and the fruits treated with OO EO had the highest SSC contents at

the end of the storage period (Figure 4C). Finally, the pH of the fruits increased during storage, however no significant effect was observed among the treatments in terms of the pH (Figure 4D). The visual appearance of the fruits and the damage caused by the *B. cinerea* is also showed in Figure 5.

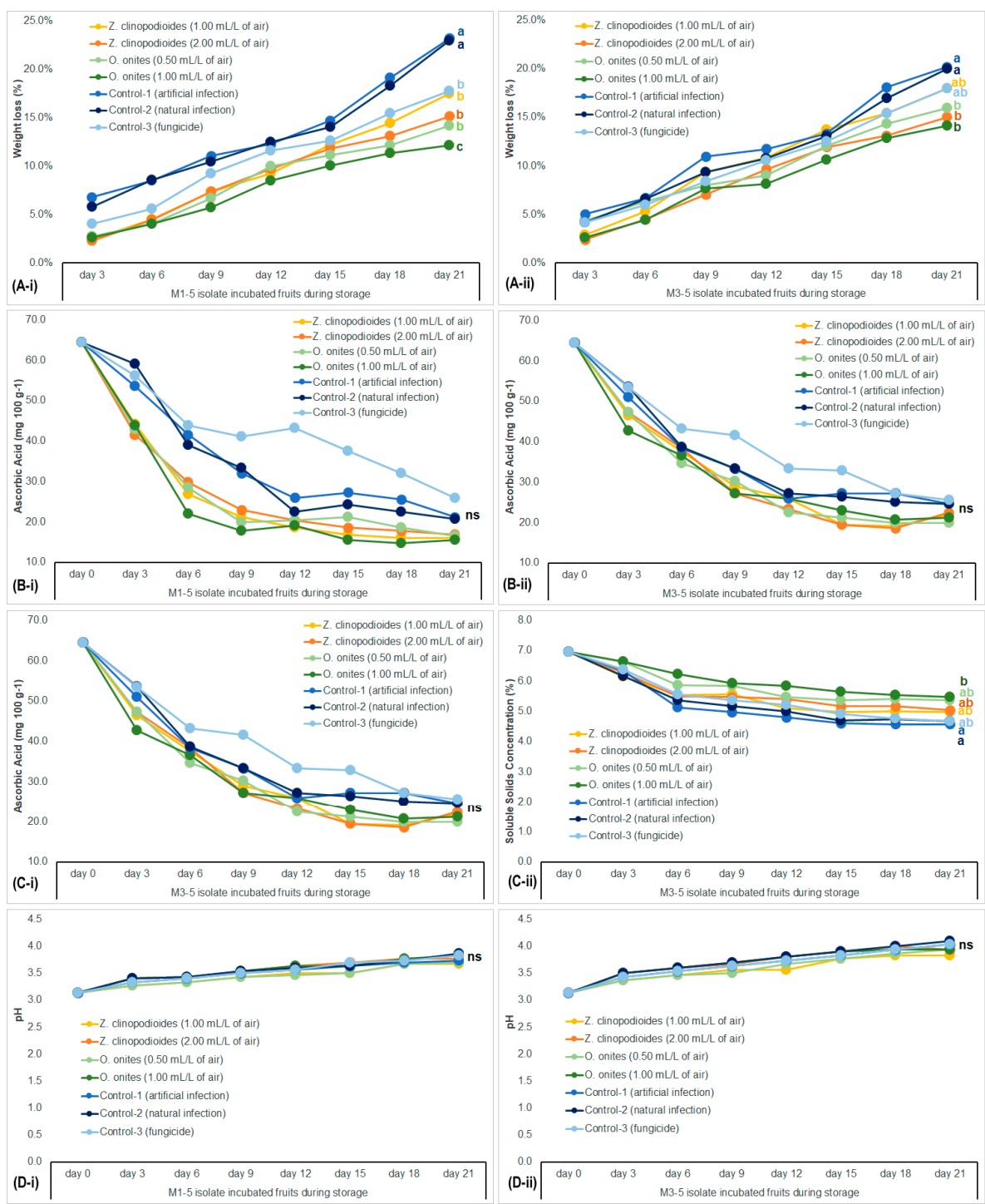

**Figure 4.** Vapor contact impacts of *O. onites* and *Z. clinopodioides* essential oils on the (**A**) weight loss, (**B**) ascorbic acid, (**C**) soluble solids concentration and (**D**) pH of strawberry fruits inoculated with (i) M1-5 and (ii) M3-5 isolates of *B. cinerea* during 21 days of storage. Different letters next to the lines of each treatment represents significant difference and ns means non-significant at the 21st day of storage, according to Tukey's HSD test ($p \leq 0.05$).

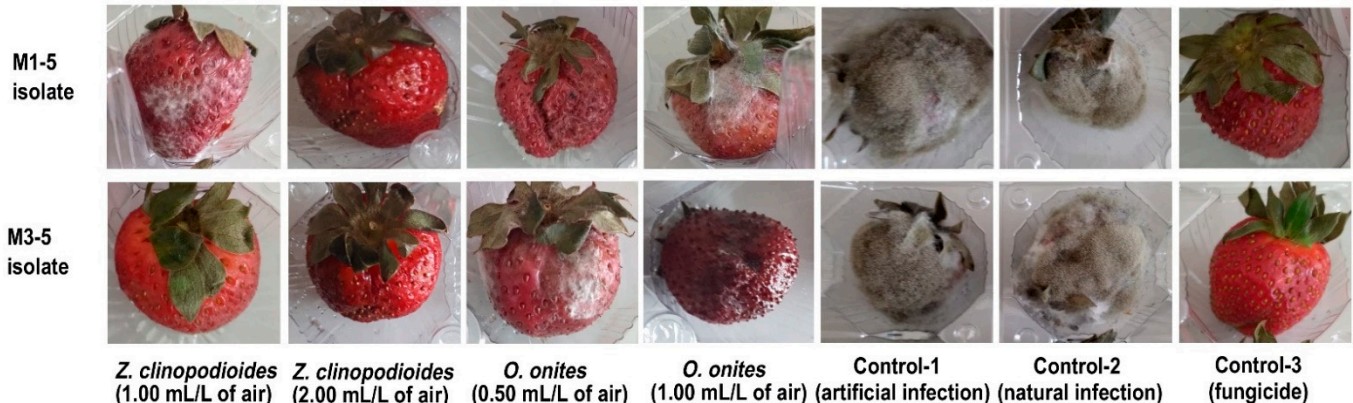

**Figure 5.** Visual appearance of fruits and disease severity at the 21st day of storage.

## 4. Discussion

The experiments showed that both *O. onites* and *Z. clinopodioides* essential oils have a significant influence on the prevention of mycelial growth under in vitro conditions. The 1.00 and 2.00 mL/L doses of OO were found to prevent the mycelial growth of all isolates of *B. cinerea*. The ZC essential oil was also noted to have a significant influence but less effect than the OO essential oil. This mycelial growth inhibition ability of the OO and ZC may be associated with their chemical compositions. The current results are in agreement with the reports of Korukluoglu et al. [29], who suggested that essential oils of *Origanum onites* have high antifungal ability against *Alternaria alternata, Aspergillus niger, A. parasiticus, Fusarium oxysporum, Mucor racemosus* and *Penicillium roqueforti*. In a similar study, Ocak et al. [30] reported that the different species of *Origanum* (*O. hypericifolium*) also have high antifungal activity. They reported an excellent performance (100%) against the mycelial growth of *Penicillium castellonense, P. verrucosum, Chatomium globosum* and *Aspergillus kiliense* on hazelnut and walnut fruits. In terms of the second EO of the current study, the *Z. clinopodioides* essential oil was also reported to be rich in chemical composition and its incorporation into chitosan coating was suggested to improve the film forming and antibacterial ability of the coating [42]. In another study, contact and vapor applications of ZC were noted by Ma et al. [26] to have antifungal activity against and prevent the mycelial growth of *Sclerotinia sclerotiorum*. They also noted that the application dose is crucial for the impact of ZC while the lower doses are not as effective, which in agreement with current results.

Vapor contact application of *O. onites* and *Z. clinopodioides* essential oils showed high performance in the prevention of disease severity on fruits in in vivo studies too. In a similar study, Mohammadi et al. [43] noted that the decay of strawberry fruits began after four days of storage at 4 °C in untreated fruits, while it was delayed by seven to nine days at *Zataria multiflora* essential oil treated fruits. Kahramanoğlu [20] reported that cv. Camarosa is a sensitive cultivar compared with cv. Sahara, cv. Florida Fortuna and cv. Rubygem, which is mostly affected by decay and weight loss. In that study, the storability of cv. Camarosa strawberry fruits was extended to 12 days with the combined application of lemongrass essential oil and modified atmosphere packaging. In agreement with this literature, the results of the current study showed that both doses of ZC and higher doses of OO have very good control of gray mold in 12 days of storage.

The antifungal activity of essential oils has been widely studied [22–26], whereas very limited information is available about the antifungal and/or antibacterial effects of OO and ZC [29,42]. In relation to the exact mechanism behind this antifungal activity of EOs, there are several proposed mechanisms. Such mechanisms include cell membrane disruption and/or expansion in fungi, loss of cell wall integrity, blocking of fungi cell growth, prevention of respiration, and increasing permeability to ions [23,44,45].

The present study showed that OO and ZC have dose dependent impacts on the prevention of disease severity caused by *B. cinerea* on strawberry fruits. Contrary to their antifungal activity and high potential as alternative to fungicides, the EOs are not water soluble and are prone to volatilization. Although there are several studies with EO incorporated edible films/coatings [46,47], these two characteristics of the EOs limit their utility in the postharvest industry for preserving fruits [48]. Moreover, it was also reported that direct EO applications have a significant impact on the sensory quality of fruit, as they alter the flavor and/or odor. Vapor contact instead of direct contact is an alternative for reducing this change in sensory quality [22]. Besides this, there are several studies where no difference was reported between the direct contact and vapor contact of EOs or vapor contact was recommended as a valuable tool for controlling fungal infections [22,34,43,49,50]. In one of these very recent and similar studies, de Oliveira Filho et al. [36] reported that *Mentha piperita, Cymbopogon martini* and *Cinnamomum camphora* have a high potential for controlling *B. cinerea* in in vitro studies and in in vivo vapor treatments on strawberry fruits. The OO and ZC, which were investigated in the current study, has some common compounds (carvacrol, *p*-cymene, pulegone, menthone, eucalyptol, limonene, thymol, etc. [27,28]) with the EOs tested by de Oliveira Filho et al. [36]. Moreover, Pedrotti et al. [51] reported that the EOs of *Baccharis dracunculifolia* with high contents of limonene provide antifungal activity against *B. cinerea* in grapes. In another similar study, Vitoratos et al. [52] investigated the antifungal effects of *Origanum vulgare, T. vulgaris* and *Citrus limon* against *B. cinerea*. They reported that all of the EOs have a moderate to high mycelial growth inhibition effect both in in vitro tests with the food poisoning technique and in in vivo tests with vapor treatment on artificially *B. cinerea* infected strawberry, tomato and cucumber fruits. In another study, Hosseini et al. [53] investigated the effects of *Allium sativum* and *Rosmarinus officinalis* EOs against anthracnose (*Colletotrichum nymphaeae*) in strawberry fruits. Researchers had undertaken these studies in both in vitro and in vivo conditions, by both contact and vapor assays. They reported that both oils, under all conditions, have a significant influence on the prevention of anthracnose. In a different study, the vapor treated EOs of *Melaleuca alternifolia* had been reported to damage the cell membrane of brown rot (*Monilinia fructicola*) in peaches and help to control the disease. All of these results showed that the vapor application of EOs provides similar effects with the direct contact application. Consequently, the results of the current study demonstrated that the OO and ZC EOs have high antifungal activity against *B. cinerea* and can be used in the postharvest handling of strawberries as a vapor treatment. In addition to preventing postharvest pathogens, maintaining storage quality of fruits is also crucial in postharvest studies. Modified atmosphere packaging (MAP) is a well-known technique for preserving fruits' storability by regulating the inner air composition and reducing the respiration rate of the fruits [54]. Therefore, further clarification may be required in regard to assessing the combined efficacy of EOs with MAP. In such a study, Mpho et al. [55] suggested that incorporating lemongrass oil (100 μL) into Whatman filter paper and introducing this into the MAP inhibited the growth of anthracnose (*C. gloeosporioides*) and preserved the quality of avocado fruits.

The OO and ZC EOs were found to prevent the weight loss of the fruits during storage. This is in agreement with the results of Khalifa et al. [46], who noted that the lemongrass oil incorporating edible coatings with chitosan provides better performance in preventing weight loss in strawberry fruits. Moreover, the ascorbic acid content of the fruits had a declining trend during storage. Atress et al. [56] similarly reported that the AA content of strawberry fruits decrease during storage. Besides the prevention of weight loss, the SSC contents of the treated fruits were also found to be higher during storage of the test fruits. Previous studies recommended that the prevention of weight loss can also help to reduce or delay the loss in SSC, which was associated with the breakdown of carbohydrates [57].

The EOs of different plants have been reported to induce some defense related antioxidant enzymes (i.e., superoxide dismutases-SOD, peroxidase-POD, catalase-CAT, polyphenol oxidase-PPO, etc.), retard ethylene production in fruits, reduce the activity of free

radicals oxidation, delay/prevent biochemical changes in fruit quality, reduce weight loss due to the formation of a protective biomembrane on the surface of the fruit (which is formed by the target components of biological products), and prevent enzymatic degradation [13,15–17,58,59]. Most of these beneficial effects of EOs were noted for direct contact application. However, vapor application of EOs was also reported to be effective. As such, Sellamuthu et al. [58] reported that thyme oil vapor enhance antioxidant enzyme activities and prevent the anthracnose development in avocado fruits, while Shao et al. [60] noted similar enzymatic activities for tea tree oil vapors which resulted in a mycelial growth inhibition of *B. cinerea* in strawberries. These enzymes do not only have defense-related roles in fruits, but also help to preserve the postharvest quality of fruits [61]. Therefore, vapor application of OO and ZC EOs can also be used for maintaining the quality of strawberry fruits.

## 5. Conclusions

The presented studies in vitro and in vivo showed the perspective of using biological preparations against microbiological spoilage during the storage of strawberries. *O. onites* and *Z. clinopodioides* essential oils have antifungal activity against *B. cinerea*. Results also demonstrated that the vapor contact application of these essential oils (1.00 mL/L of air application dose *O. onites* and 2.00 mL/L of air application dose for *Z. clinopodioides*) serve as an alternative to fungicides for controlling postharvest gray mold of strawberry fruits (cv. Camarosa) caused by *B. cinerea*. Further studies are needed for the combined efficacy of volatile essential oils of *O. onites* and *Z. clinopodioides* with packaging materials (such as modified atmosphere packaging) and the impact of phytopathogenic microorganisms on strawberry cultivars (including fruit anatomy). Revealing their combined efficacy would help to develop industrial materials (complex biological product) for the postharvest handling of fruits. It is also necessary to test the sensorial impacts of the EOs on the fruits before considering them as an alternative method in postharvest fruit storage.

**Author Contributions:** Conceptualization, İ.K., O.P., T.G.K., A.U.B., R.G. and H.A.; methodology, İ.K., O.P., T.G.K. and A.U.B.; validation, İ.K., O.P. and R.G.; investigation, T.G.K., A.U.B., R.G. and H.A.; resources, T.G.K., A.U.B., R.G. and H.A.; data curation, İ.K. and O.P.; writing-original draft preparation, İ.K.; writing-review and editing, O.P., T.G.K., A.U.B., R.G. and H.A.; visualization, İ.K.; funding acquisition, O.P. All authors have read and agreed to the published version of the manuscript.

**Funding:** Ministry of Science and Higher Education of the Russian Federation project FGZS-2022-0007.

**Institutional Review Board Statement:** Not applicable.

**Informed Consent Statement:** Not applicable.

**Data Availability Statement:** All data is presented in the paper.

**Conflicts of Interest:** The authors declare no conflict of interest.

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
