# Peer review of "Control of Postharvest Gray Mold at Strawberry Fruits Caused by Botrytis cinerea and Improving Fruit Storability through Origanum onites L. and Ziziphora clinopodioides L. Volatile Essential Oils"

_agronomy, doi:10.3390/agronomy12020389_

Round 1
Reviewer 1 Report
suggest title: Control of postharvest gray mold at strawberry fruits caused by Botrytis cinerea and improving fruits storability through Origanum onites L. and Ziziphora clinopodioides L. essential oils.
and decide gray or grey mold. now in some text grey in other gray. in title gray.
line 17: Botrytis cinerea italic.
in whole text B. cinerea italic.
in vitro and in vivo italic
25 line: suggestion change to: essential oils have a moderate-to-high impact on the prevention of grey mold....
26 line : correct to : oils were also found to have slight-to-moderate impact on weight loss and loss of ...
88 line: BBCH
88 line: so you made essential oils from a location mix of materials? or was extracted one essential oil from one location?
94 line: freezer at +4 °C or -4 °C
97 line: question monocultures? identified by PCR? origin?
106 line: why only ethanol was chosen? why not tween?
why 25 C temperature chosen? Botrytis better grows on 22 C
123 line: BBCH fruits, were fruits treated fungicides in the field (routine applications)????
126 line: temperature
104 line: Euloge et al. [32 not talks about poisoned food technique. and in Petri should be called direct effect in Petri , its not on food. its on fungi.
115 line: replicated how many times???
130 line: question why these concentrations? I assume according to results. but please explain so. and why only 2 isolates of 3 were used in vivo.
137 line: concentration 106?????? it should be 10 6
137 line: not infected but injected?
136 line: what scalpel is 10 mL, mayeby pipet tip?
138 line: in boxes on paper or direct in the box? shape and size of the box. how many fruits per box?
I find a lot of info below, but in front, it raises the question, so please structurize.
139 line : upper you write that different doses for each essential oil, here, I understand 3 doses per one essential oil?
142 line: paper plate? or filter paper?
142: line: how? inner part of the box cover? I understand boxes were without cover.
149: why not sprayed on fruits like with fungicide?
150: EPPO PP1/016 say If desired, fruit may be assessed after 2–5 days of simulated storage at 5–7°C.
so I don see why 25 is a regular temperature.
how many fruits per replicate and treatment?
152: how fruits looked after 21 day of inoculation?????
160: methods of soluble solids concentration, pH and 160 ascorbic acid content? reference?
164: refractomer - provider?
165: pH meter. - same , provider
168: anova?????
176: italic O. onites and Z. clinopodioides
177: not in methods direct contact method. this info in methods should be: (colony 177 diameter - cm)
181: use abbreviations sterile water and 70% ethanol, and write them in methods. ex: control-1, control-2.... like in table 1 . but this should be in methods at first
Figure 1. not clear where Fig. 1 A and where 1B, make text size readable.
fig. 2 . text size readable, now hard to read, too small
fig 3. in methods there is no info about 1,00 ml/l of air. write 1,00 ml/l its enough. also it axis enough control-1, after the title of figure please write what means control-1. but if figure its not necessary
217: not clear why these concentrations choose, write sentence why
figure 4: better table, easier to read. and fig 4 should be near the data.
did you made chemotypes of your essential oils?
you write 269 line about volatile compositions, but not of your used essential oils.
273: italic Z. clinopodioides
367: which concentrations recommend to use?
202: what EOs means? I understand essential oil, o use in all text EOs, now previously were essential oils, from 202 line EOs
Author Response
We have made the appropriate corrections, which we hope you will approve. Responses to comments are provided below (file).

Reviewer 2 Report
This research provides a very interesting approach to solving storage diseases in strawberry production, especially with the use of biological agents.
Please, correct and more clearly describe the material and methods of manuscript.
Author Response
We have made corresponding corrections that we hope will meet with your approval. The responses to the comments are provided below (file).

Reviewer 3 Report
This work deals with a hot topic in the context of the biocontrol strategies, studying by in vitro test the antifungal activities of Origanum onites L. and Ziziphora clinopodioides L. essential oils against three different isolates of Botrytis cinerea and by in vivo test the vapor contact impacts on fungus and strawberry fruit quality. The identification of new potential alternatives to fungicides is here discussed. Therefore, analysis for the determination of O. onites and Z. clinopodioides efficacy against B. cinerea was conducted in in vitro studies and their effects on the storability of strawberry fruits cv. Camarosa were tested under in vivo conditions. The main conclusion is that the tested EOs could be potential alternative to fungicides.
Major comments:
- I think it is necessary to investigate the chemical composition of the EOs (as described in Tyagi & Malik, 2011 “Antimicrobial potential and chemical composition of Eucalyptus globulus oil in liquid and vapour phase against food spoilage microorganisms”), in order to determine the p-cymene, thymol and carvacrol contents because of their possible toxicity for human health due to high dose consumption (as described for p-cymene in Fraternale et al. 2013 “Cytotoxic Activity of Essential Oils of Aerial Parts and Ripe Fruits of Echinophora spinosa (Apiaceae)”).
- For the above cited reason, I think it could be necessary to investigate what is the residual content of p-cymene, thymol and carvacrol on fruit peel after washing processes.
- I think it could be necessary to perform a sensorial analysis to evaluate the EOs effect on the strawberry taste (as described in Espina et al. 2014 “Impact of Essential Oils on the Taste Acceptance of Tomato Juice, Vegetable Soup, or Poultry Burgers”).
Minor comments:
- I suggest improving the background related to postharvest fruits and cinerea using the most recent review about biocontrol “The Role of Yeasts as Biocontrol Agents for Pathogenic Fungi on Postharvest Grapes: A Review” (Di Canito et al. 2021).
- Several times the name of the organisms in the text is not in Italic form, I suggest checking them in the whole text (also in Tables and Graphs).
- Please modify “p-cymene” with “p-cymene”.
- “In vivo” and “in vitro” must be written in Italic form. Please modify them.
- The symbol “&” is a misprint in several part of the text. Please remove.
- Line 94: “The essential oils were stored in a freezer at 4 °C until they used in…” I think you must modify freezer with refrigerator.
- Line 137: “1.0 × 106 conidia/mL” must be modified in “1.0 × 106 conidia/mL”
- Line 182: “This made it possible to conclude that the 70% ethanol, had no significant influence on the mycelial growth.” The underlined “the” must be omitted.
- I suggest checking the form of the references, for example in some cases the year of publishing is not in bold form.
- I recommend sending the text to an English editing service prior to resubmission.
Author Response

(The authors gave the same response as above.)

Round 2
Reviewer 1 Report
108: question you don't evaluate chemotypes, but in your colleagues from the same plant material extracted EO and they evaluated chemotypes? if its not the same essential oil its not needs to add table 1, because in different plant materials could be a bit different chemotypes composition, it depends on many factors.
152: so only M2-1 were PCR confirmed B. cinerea?
nor understandable: M2-1 coded isolate was selected from the 3 isolates, and DNA sequencing was performed by sequencing.
I suggest: M2-1 isolate DNA identified by sequencing (Reference)
153: why 25 C temperature?
157: what modifications, please describe the modifications
183: how were sprayed strawberries during growth? or during vegetation, no pesticides were used on plants???
213: why here is the aim? the aim should be in the end of the introduction part

Author Response
Dear reviewer
We answered on your questions and made corrections to the manuscript.
With respect,
Reviewer 1:
"I don't understand why the authors give chemotype compositions from other studies if they don't evaluate their extracted essential oils. I understand if it were the same plants, the same growth period and time, but if it's literature, then the method table is not needed ".
We indicated that the biochemical analysis of the oil was not studied in this study (lines 94-95). This not was the aim of this study. In future studies, we will definitely take into account the recommendations of the reviewer and make a chemical analysis of the oil.
Following the comments of Reviewer #1, we think it would be useful to include some information on the composition of oils from other similar studies conducted in similar climates. That is why we have placed this table 1 in the first edition. We have now removed it from the revised document. 1. So, lines 135-140 are not methods and are not needed.
176 line: Inhibition of mycelial growth is it the same mycelial growth Inhibition as in fig. 2? Dear authors, please use the same meanings.
Thanks to the reviewer, we corrected the sentence in line 176 and ensured the same meaning.
Figures 1 and 3, 4: should be improved DPI.
The qualities of the original figures are high. However, their quality is reduced in the system. As we mentioned in our first revision, in case of need, we will improve their quality during the production of the paper. Thank you for the comment.
108: question you don't evaluate chemotypes, but in your colleagues from the same plant material extracted EO and they evaluated chemotypes? If it’s not the same essential oil it’s not needs to add table 1, because in different plant materials could be a bit different chemotypes composition, it depends on many factors.
We deleted table 1
152: so only M2-1 was PCR confirmed B. cinerea?
Nor understandable: M2-1 coded isolate was selected from the 3 isolates, and DNA sequencing was performed by sequencing.
I suggest: M2-1 isolate DNA identified by sequencing (Reference)
We edited this section in the methodology
The three isolates of Botrytis cinerea, M1-5, M2-1 and M3-5, were supplied from the culture collection of Iğdır University, Faculty of Agriculture, Department of Plant Protec-tion and Phytopathology Laboratory. The isolates were identified microscopically ac-cording to the features of mycelium such as appearance and color, as well as conidia, co-nidiophore and sclerotia. M2-1 isolate DNA identified by sequencing .
153: why 25 C temperature?
Temperature +25 C used according to recommendations "Mycology. Experimental learning methods Microscopic mushrooms “
157: what modifications, please describe the modifications
We wrote which modifications we did. Line 124-140.
183: how were sprayed strawberries during growth? or during vegetation, no pesticides were used on plants???
During budding and flowering, nitrogen, potash and phosphorus fertilizers were used. During the fruiting period, mineral fertilizers were not used. Pesticides have not been used.
213: why here is the aim? The aim should be in the end of the introduction part
This is an explanation: Current research has focused on testing the effects of B. cinerea EO steam contact on strawberry fruit, not just B. cinerea.
Reviewer 3 Report
I think that your explanations and revisions are sufficient to accept your paper.
I suggest to perform an English language editing.
Author Response
Dear reviewer!
We uploaded the corrected version of the manuscript to the system